# Dynamic Macronutrient Meal-Equivalent Menu Method: Towards Individual Nutrition Intervention Programs

**DOI:** 10.3390/mps2030078

**Published:** 2019-09-05

**Authors:** Ana Teresa Limon-Miro, Veronica Lopez-Teros, Humberto Astiazaran-Garcia

**Affiliations:** 1Department of Nutrition, Centro de Investigacion en Alimentacion y Desarrollo A.C. (CIAD), Carretera Gustavo Enrique Astiazaran Rosas 46, Hermosillo 83304, Sonora, Mexico; 2Department of Chemical and Biological Sciences, Universidad de Sonora, Hermosillo 83000, Sonora, Mexico

**Keywords:** nutrition care process, nutrition counseling, dietary treatment adherence, dietetics, dietary assessment, ambulatory nutrition service, clinical nutrition management

## Abstract

Nutrition interventions should reflect the state of the art in science and dietetics to meet each patient’s requirements. Incorporating new knowledge into individualized food-based nutrition interventions is a major challenge, and health care professionals constantly search for novel approaches through specific and standardized methods. The dynamic macronutrient meal-equivalent menu method involves individuals making informed food choices that match their requirements, schedule, and food availability and affinity, helping them maintain a sense of control and motivation to adhere to a nutrition intervention program. This protocol includes the steps required to prepare a nutrition plan containing equivalent meal options consistent with the patient’s needs and preferences. Standard food servings are planned according to population specific dietary guidelines and individual characteristics. Servings are distributed at required mealtimes, and are all equivalent in energy and macronutrient content, providing every patient with interchangeable choices within each mealtime. This empowers individuals to select foods in a guided format whilst adhering to a dietary plan. Acceptable variations for calculated energy and macronutrient content are as follows: protein ±1 g/day, fat ±1 g/day, carbohydrate ±2 g/day, and energy ±15 kcal/day. Following this method, health care professionals can develop individualized nutrition intervention programs that may improve patients’ adherence, nutritional status, and health.

## 1. Introduction

According to the United Nations Food and Agriculture Organization and World Health Organization, food-based dietary guidelines (FBDGs) state the principles of food-based nutrition education to guide the population, food industry, and national nutrition policies. Countries have developed their guidelines according to the population’s dietary intake patterns, availability, access, and food composition, as well as their epidemiological profile [1,2,3]. 

Assessing compliance to national FBDGs [3] at the population level is uncommon, and although limited evidence is available from some countries, the observed feeding patterns are not in agreement with their FBDGs [4]. Overall, population dietary patterns exceed sugar, salt, and fat intake, and at the same time fail to meet fruit, vegetable, and fish adequate consumption [3,4]. Hence, emerging FBDGs and individualized dietary approaches should provide tools for health care professionals (HCPs), especially registered dietitian nutritionists (RDNs), to facilitate dietary planning during the nutrition care process (NCP) in order to meet the patient’s dietary reference intake [3,5]. 

Personalized or individualized nutrition assessment is an evidence-based application during the NCP, which is implemented according to the patient’s nutrition diagnosis and is tailored to manage a disease, injury, or specific condition; it is also used for intervention planning, periodic monitoring, evaluation, and re-assessment [6]. Considering that the individual’s response to a dietary intervention can be influenced by environmental and genetic factors, the need for a more personalized approach to nutrition and dietary advice is highly evident [4]. This emerging field could enable the adjustment of FBDGs for specific population needs based on their molecular profiles [7]. Yet, according to the Academy of Nutrition and Dietetics, the use of nutrigenetic testing to provide dietary advice is not ready for routine practice [7], and the Academy of the International Society of Nutrigenetics/Nutrigenomics states that many aspects in this sense remain as a challenge for the future of nutrition [8]. Furthermore, consumers’ attitude towards genetic testing include skepticism, mistrust, unproven efficacy, and disinterest [9]. Thus, tailoring an individual’s diet based on nutrigenomics (personalized nutrition) does not guarantee that a feeding behavior change will occur. Therefore, personalized nutrition interventions according to the individual’s phenotypic particularities still refers to a hypothetical concept, rather than a fact, for individuals assessed by the HCP or RDN [9,10]. 

Nutrition interventions should also consider psychological and socio-cultural factors that shape consumers’ perception, attitudes, and decision-making related to feeding behavior. Quality of life and a healthy nutritional status in adults can be enhanced by individualized nutrition approaches. The Academy of Nutrition and Dietetics states that as part of a multidisciplinary team, the RDN must evaluate each individual and assess the risks vs. benefits of a nutritional intervention, whilst developing a nutritional plan according to each individual’s nutritional status, medical condition, personal preferences, and right to make informed choices [11].

Nutrition care provided by the RDN and HCP should reflect the state of the art in both science and dietetics to meet each patient’s requirements [5]. Incorporating new knowledge into nutrition research and practice to achieve evidence-based recommendations and personalized food-based diets is a major challenge, as it must consider not only the selection of foods, but also the behavioral challenges to address adherence, planning skills, personal motivation, and financial constraints [3,4]. As a result, RDNs constantly search for novel approaches to tailor diets individually by using specific and standardized methods [5].

The dynamic macronutrient meal-equivalent menu method (MEM) involves individuals making informed food choices that match their requirements, schedule, and food availability and affinity, helping them maintain a sense of control and motivation to adhere to a nutrition intervention program. It involves a specific and systematic way to calculate energy and macronutrient content in order to adapt it to a nutrition plan that will provide equivalent meal options in macronutrient content, consistent with each person’s needs and preferences. This method can empower patients to select foods in a guided format whilst adhering to a dietary plan; moreover, it can help HCP to develop individualized nutrition interventions that can improve the patients’ adherence, nutritional status, and health.

## 2. Methodology

### 2.1. Understanding Individuals and Involving Them in the Process

The path towards a personalized nutrition method requires a better understanding of the individual’s dietary choices and his/her social, cultural, and economic background so that specific macronutrient requirements can be fulfilled [12]. An individualized dietary intervention should be discussed with the patient and take into account his/her preferences and goals to develop his/her informed options, rather than exclusively considering diagnosis [5,11]. Involving individuals in the development of the dietary plan that matches their requirements, daily schedule, and also considers food availability, access, and affinity, can create a sense of control, autonomy, and motivation that can help them maintain short- and long-term adherence [11].

Co-designing the nutrition intervention and tailoring information to influence changes towards a healthier behavior, rather than the provision of general information or recommendations, is valuable because subjects who are well informed about potential benefits associated with individualized nutrition perceive they can genuinely achieve their health goals [4]. In addition, attitude has been shown to be a reliable predictor of behavioral intention to make certain food choices, while freedom of choice has been determined as the most important promoter of acceptance. Still, behavior change is determined by perceptions of self-efficacy and individual outcome expectations, as well as the ability to change a given behavior and visualize the expected outcome. Therefore, self-efficacy has been associated with the perceived ability and intention to make healthy food choices [10], all of which are factors considered in the MEM.

### 2.2. Individualized Macronutrient Meal-Equivalent Menu Method as Part of the Nutrition Care Process

As mentioned earlier, behavior and socioeconomic factors play a role in the selection of dietary components to successfully maintain optimal health. Additional benefits of individualizing dietary plans may include reducing health care costs, not only for patients seeking nutrition counseling through the NCP but also for ambulatory nutrition services, nutrition programs, or facilities that provide food services by offering a general diet appropriate for the majority of their customers [11], as the individual’s characteristics and preferences are considered a priority. An individualized dietary plan does not mean that the patient will control the nutrition intervention, but that he/she will actively participate and learn to organize his/her meals. 

A dynamic individualized macronutrient meal-equivalent menu is a nutrition plan that will provide every patient with seven interchangeable options for each meal, all equivalent in macronutrient content, according to their individual needs and preferences. This will empower them in the selection of foods in a guided format whilst adhering to a dietary plan. Having equivalent macronutrient content in different meal options will help the patient to overcome obstacles like stress, anxiety, high-cost of nutrition plans [13,14], and isolation from family members, friends, and social reunions [15,16]. The method is intended to be applied in patients who need nutrition counseling during the NCP, and for nutrition services in ambulatory settings [11].

Multiple meal options based on the individual’s preferences and condition will allow the patient to feel guided but not restricted, and as a result, it may be easier for them to cook, afford, and enjoy the meals. Directing nutritional counseling towards an assertive nutritional guidance like the MEM could promote each patient’s short- and long-term adherence to nutrition intervention programs [5,11]. Here we propose a method to design macronutrient meal-equivalent menu that will help RDNs and HCPs to individualize nutrition interventions, reduce time in dietary planning, and enable patients to adhere to a short- and long-term nutrition protocol.

In the first session, the individual and a RDN or HCP must be present. Children and teenagers must be accompanied by at least one parent or guardian. Depending on the RDN/HCP experience, the first session could last approximately 60–90 min, and further follow-up sessions may be around 30–40 min. The MEM does not require specialized equipment, but it is best to use software with spreadsheets for energy and macronutrient calculations and a word processing program for the macronutrient meal-equivalent menu. Body composition data should be assessed with the most accurate and precise equipment available.

In order to successfully design a food-based individualized diet, information on each individual should be collected using validated tools during the NCP [5]. In-depth interviewing should be conducted with minimum bias (e.g., not influencing the patient’s answers), to be able to identify their habits and food choices. Professional feedback and self-monitoring should be continuous throughout the process to promote adherence and identify barriers or misconceptions [8,12]. The RDN must consider socioeconomic [13,17], educational, cultural [12,16], occupational, environmental [16], and emotional/mental factors [12,14], nutritional status, food preferences, physical activity, family history, food sensitivity, and biological determinants [17] (e.g., disease) as factors that influence food behavior and must be part of the NCP assessment (Figure 1) [5]. 

### 2.3. Energy and Macronutrient Distribution

To maximize the potential health benefits of food-based dietary interventions, an adequate distribution of energy and macronutrients is critical for developing an individualized diet [17,18]. Isocaloric diets with different macronutrient distribution can affect appetite, metabolism, and thermogenesis [19]. Additionally, variations in macronutrient distribution can directly affect adherence to dietary recommendations and guidelines, depending on the individual’s characteristics [17] and associated environmental factors (Figure 1) [12,16].

Based on the individual’s nutritional status and when weight loss is advised, some studies may suggest a caloric restriction (500–1000 kcal/day) [17,19,20], and although caloric intake is recognized as a critical factor for weight reduction [19], the macronutrient distribution [18] and food selection (e.g., replacing saturated fat with mono- and polyunsaturated fatty acids and low-glycemic foods) play a crucial role in long-term weight loss maintenance [21] and associated non-communicable diseases prevention and treatment [1,17].

Energy and macronutrient requirements are set depending on the individual’s age, sex, nutritional status, physical activity level, pathologies, etc. [17,18]. Once this information is complete and the caloric intake requirement has been established, the HCP can estimate the amount and distribution of dietary macronutrients to develop an individualized plan, considering FBDGs [1,2,3] and determinants discussed previously (Figure 1) [17,18].

The MEM starts by developing a summarized theoretical dietary table that includes energy, protein, carbohydrate, and total fat distribution, according to FBDGs [1,2,3,17], clinical nutrition guidelines [22,23,24], and the acceptable macronutrient distribution range [22,23,25]. Every macronutrient should be covered in amount (g/day) and energy (kcal/day) (Appendix A) [22].

Protein requirements depend on the individual’s health status and physical activity level; hence, there is a wide range of protein intake (0.75–2 g/kg BW/day). Using grams per kilogram of body weight per day (g/kg BW/day) to estimate protein intake will maintain the nitrogen balance and may prevent skeletal muscle catabolism [26]. Thus, we strongly recommend that the protein content be calculated prior to carbohydrates or fats, and based on g/kg BW/day instead of energy-macronutrient percentages [22,26]. Fats can be added next into the plan, as their recommended proportion is generally lower than that of carbohydrates, but amounts can vary depending on the patient’s clinical diagnosis [22,23,24]. Once protein and fat content has been considered, carbohydrate intake is easier to assess [1,22].

Once SFSs for every food group have been established for the patient, the obtained macronutrient content (g/day) must be compared with the theoretical assumptions, and should preferably be displayed in a comparative dietary summarized table [22]. Acceptable variations to guarantee that calculated SFSs meet the theoretical dietary intake estimates are as follows: protein ±1 g/day, fat ±1 g/day, carbohydrates ±2 g/day, and energy ±15 kcal/day.

## 3. Analysis

A qualitative analysis is necessary to incorporate habitual food choices and meals matching them with the obtained SFS/day for each individual in the MEM energy and macronutrient calculations. Given that the habitual food intake of each individual can be singularly diverse, in-depth interviewing is a useful qualitative approach to identify actual dietary practices and describe facilitators and barriers to accomplish the recommended feeding practices for each person according to FBDGs [27].

When the theoretical calculations of macronutrients intake are completed, FBDGs and personal determinants (Figure 1) should be considered again [1,2,3] to plan the number of standard food servings (SFSs) by food group which will be the foundation to develop the macronutrient meal-equivalent menu. Each SFS is equivalent in energy and macronutrients and is defined in household measures [1,25,28].

### Mealtime Planning

Calculated SFS/day should be within the acceptable variation described above and distributed in daily meals according to the patient’s needs, lifestyle, schedule, and habitual meals (Appendix A) [17,18]. Before the macronutrient meal-equivalent menu is designed, the HCP/RDN must have established the meal times and schedule that the patient will have, preventing prolonged fasting periods [29] (>5 h) and in agreement with the patient, so that it is feasible for him/her to follow the schedule. Each mealtime should have the same number of SFSs. Therefore, the HCP and RDN guidance can help organize the patient’s schedule, maintaining, adding, or deleting specific mealtimes, to promote and sustain a healthy mealtime schedule. This can also guarantee that the patient’s gastric capacity and digestion will not be significantly altered [15].

Next, the facilitator must do his/her best to adapt the meal options to the patient’s regular intake. The HCP/RDN should pay special attention to the individual’s favorite meals and recipes, trying to adjust them to the calculated SFSs per meal. Patients should be advised in advance about the adjustments in portion sizes and be encouraged to use household measures to incorporate food groups in the quantities specified in the nutrition plan.

Those new meals and recipes included should match what the patient regularly buys in the supermarket. Furthermore, according to his/her socioeconomic level and culinary techniques, the new options provided should agree with all of these factors (Figure 1). It is recommended for the HCP/RDN to verify, once a new recipe or meal option is considered, whether the patient is willing to cook it at home and if he/she understands all the procedures and ingredients needed. If a certain ingredient is unknown or unaffordable, the HCP/RDN must provide an alternative that the patient could consume or discuss if changing to a new meal option or recipe would be best for him/her.

## 4. Expected Result

### 4.1. Dynamic Macronutrient Meal-Equivalent Menu

In the MEM, the expected result is for the patient to have seven interchangeable options for each meal, all equivalent in SFSs and hence in energy and macronutrient content, consolidated in a menu (Figure 2).

An example of a dynamic macronutrient meal-equivalent menu is represented in Appendix A. In order to maintain adequate daily macronutrients and energy intake, the HCP must include every food group and calculated SFSs at each mealtime and respect them in all the options for every planned meal. This format and method empowers the patient to make meal-choices within each mealtime in a guided format whilst adhering to a dietary plan (Figure 3).

SFSs are translated into particular food combinations and recommendations, organized by columns and rows, where each column contains the different meal options, and the rows contain the mealtimes including the planned schedule, providing every patient with interchangeable choices within each mealtime, all equivalent in energy and macronutrient content (Figure 2 and Figure 3). Therefore, the SFSs resemble the exchange patterns for calculated diets using population-specific food exchange lists [22,25,28].

### 4.2. Follow-up Sessions 

Professional feedback and self-monitoring should be constant throughout the intervention to enhance adherence and identify barriers or misconceptions [8,12,18]. Follow-up visits and new additional 7-day menu options should be considered weekly or every 2 weeks, until the nutrition intervention goals are met and sustained.

During follow-up sessions, the HCP should review each mealtime and meal option, removing those that did not work out for the patient (e.g., in a financial, psychological, and/or practical way). When the nutritional assessment shows that the patient needs to maintain the same energy and macronutrient intake, as well as the meal distribution, the HCP can include new meal options respecting the theoretical calculations and the individual’s particularities. On the other hand, if the patient’s conditions change during the intervention and the nutrition plan requires adjustments in energy and macronutrient distribution, all calculations should be redone. Before suggesting any new meal options, the patient’s opinion must be taken into account, checking if he/she would like to include something that can fit in his/her meal options, access and availability to certain foods, and food preferences, which can create a sense of control, autonomy, and motivation that can help maintain short- and long-term adherence [11]. 

### 4.3. Summarizing the MEM steps 

Assess the nutritional status and physical activity level to calculate the theoretical energy and macronutrients.Adjustments in daily caloric intake (restriction/addition) should be made considering the individual’s health status.The macronutrients intake should be calculated in the following order: protein (g/kg BW/day), fat, and carbohydrates.The SFSs/day/food group are set according to the patient preferences, socioeconomic and cultural background, as well as theoretical energy and macronutrient calculations.Once SFS/day have been established, the obtained energy (kcal/day) and macronutrient content (g/day) must be compared with the theoretical estimates within acceptable variations: protein ±1 g/day, fat ± 1 g/day, carbohydrate ± 2 g/day, and energy ± 15 kcal/day.Based on the individual schedule and a healthy mealtime distribution, the RDN plans the required mealtimes.Total SFS/day are distributed within the planned schedule. Therefore, each meal will be equivalent in energy and macronutrient content.The meal options will be designed depending on the patient’s access to food and regular meals maintaining the SFS/meal.Seven interchangeable meal options within each mealtime will be provided by the RDN, enabling patients to make healthy meal-choices in a guided format whilst adhering to a dietary plan.During the follow-up sessions, the RDN will evaluate how the mealtimes and food options worked for the patient, making adaptations for the next period.

## 5. Discussion

Individualized nutrition and lifestyle programs are more effective than standard care [30] or universal dietary recommendations [14], especially when combined with professional support and self-monitoring [18,30]. Standard care refers to the minimum level of care that should be received by all patients, regardless of their nutritional status, and sometimes can include a standard diet and physical activity advice [30].

The MEM offers a practical approach to individualized nutrition intervention programs. This method allows the HCP to adapt the dietary recommendations to the patient’s preferences taking into account their nutritional status, health needs, and individual and environmental determinants, giving patients a sense of autonomy and empowerment, thus helping them to achieve their health goals by making adequate food choices. 

Using this approach and after an adaptation period, patients could be more open to include new food sources and combinations by realizing that healthy eating can be easier than they expected. If the HCP begins the nutrition intervention planning by considering the patient’s preferred foods and eating habits, this will provide patients with a sense of knowledge and experience of eating nutrient-rich foods and food groups, thereby encouraging them to progressively include higher-value foods in their diet [1,7,22,23,24]. The planning formats to elaborate a dynamic macronutrient meal-equivalent menu method are included in Appendix A.

Individualized nutrition interventions may improve adherence to the nutrition treatment. In a study where patients followed a non-personalized prescribed diet, the non-adherence rates to the prescribed diet at 3, 6, 9, and 12 months were 39%, 45%, 51%, and 74%, respectively. The main barriers for adherence were poor self-discipline (72%), low family support (11%), and diet cost (20%) [15].

Poor self-discipline can refer to a broad range of factors associated with the patient’s perception of the nutrition plan as restrictive, limited, and contrary to his/her personal preferences. An individualized nutrition plan such as the MEM can address this issue because from the beginning of the intervention, it guarantees an adequate energy and macronutrients content, while also considering the individual’s preferred and common meals. However, it is likely that experienced RDNs are needed to implement this method, as it requires practice, patience, and time to build a rapport and empathic bond with the patient in order to ascertain the obvious and subtle characteristics and preferences of each individual [5,17]. 

A limitation that patients frequently state when they are not completely willing to follow a dietary plan is that they associate higher costs with therapeutic diets and/or healthier eating patterns [13], which is not necessarily true. Nutrition interventions should be designed considering the patient’s socioeconomic status in order to guarantee high-quality and affordable nutritious food sources. Considering that the MEM includes the most common meals and food sources for patients and their families, diet costs can theoretically be controlled because they will be in accordance with their usual household expenditure on food. As mentioned before, it is important that the HCP and RDN have sufficient experience, knowledge, and empathy to adapt the MEM to the patient’s individual characteristics, and are aware of seasonal foods, and food availability and prices at local supermarkets and grocery stores to be able to guide each patient. Moreover, the involvement of the family will improve the patient’s long-term adherence and eating habits; therefore, it should be encouraged and considered [16,31].

As described above, the exchange food lists [25,26,28] provide the fundamentals for the conception of the MEM, which aims to guarantee a more dynamic and independent way for each patient to choose healthy food meals according to their personal characteristics and health goals [32]. Thus, the macronutrients intake and selection of foods should be specifically tailored for each patient following current FBDGs and considering the patients’ diagnosis and nutritional status when elaborating a nutrition plan [5,17,18].

Diet personalization has a positive effect not only on non-communicable diseases prevention and treatment, but also on quality of life, increasing the effectiveness of diet therapy [5,17,18]. The implementation of an individualized nutrition program relies on the shared responsibility of the HCP and the patient, who will be willing to change his/her feeding patterns and behavior, rather than solely on the execution of the HCP or RDN [3]. In this way, standardized methods such as the MEM can be an excellent tool for professionals. In general, RDNs play a primary role in the design of nutrition programs to improve and maintain patients’ health [5,32]. Likewise, this approach will aid both HCPs and RDNs to plan individualized dietary interventions, improving the patient’s adherence, particularly for special conditions and nutrient requirements [14,15,32].

## 6. Conclusions

By using individualized nutrition intervention methods we can narrow the gap to improve health and non-communicable disease prevention in the short-term, empowering individuals to make healthy dietary choices according to their preferred foods and characteristics. This method can help health care professionals and RDNs to develop individualized nutrition interventions that can improve patients’ adherence, nutritional status, and health. Individualized food-based interventions can be challenging for the HCP/RDN because they must incorporate personal preferences, physical activity, and socioeconomic and environmental determinants, which are the main variables that influence health outcomes.

The precision of the dynamic macronutrient meal-equivalent menu method presents an opportunity for HCPs/RDNs to use it in individualized food-based interventions for patients with special nutritional requirements (pregnancy, lactation, sports, clinical nutrition, etc.).

Future research should focus on adherence, cost-effectiveness, and personal and environmental barriers in this and other methods, to help HCPs and RDNs to develop personalized nutrition dietary strategies in the NCP to ensure patients’ maximum short- and long-term benefits.

## Figures and Tables

**Figure 1 mps-02-00078-f001:**
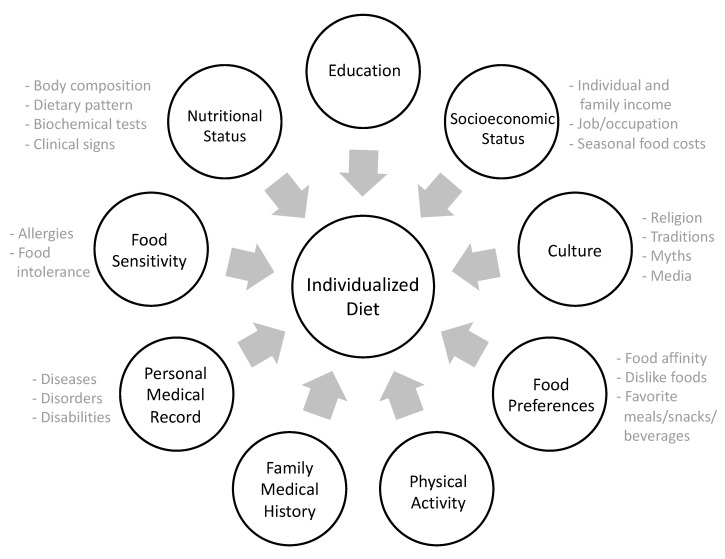
Interrelation of individual and environmental determinants considered to design individualized dietary interventions.

**Figure 2 mps-02-00078-f002:**
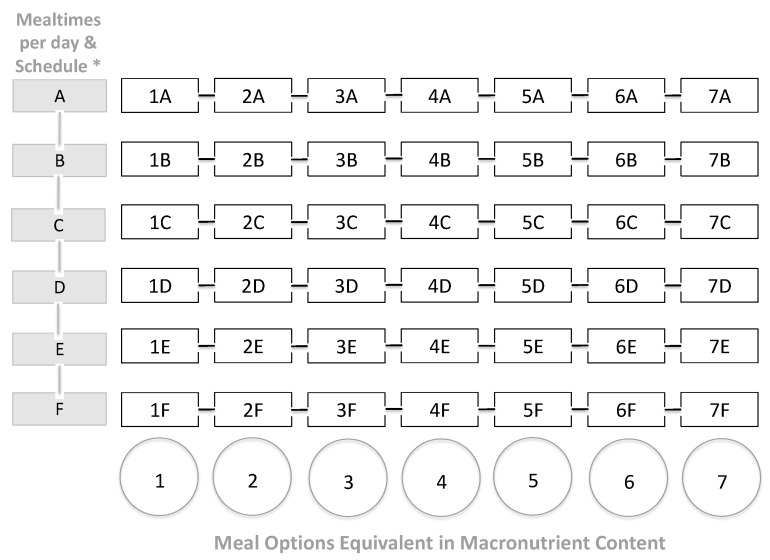
Individualized macronutrient meal-equivalent menu model. *The health care professional/registered dietitian nutritionist coordinated with the patient will set the mealtimes and schedule for the MEM.

**Figure 3 mps-02-00078-f003:**
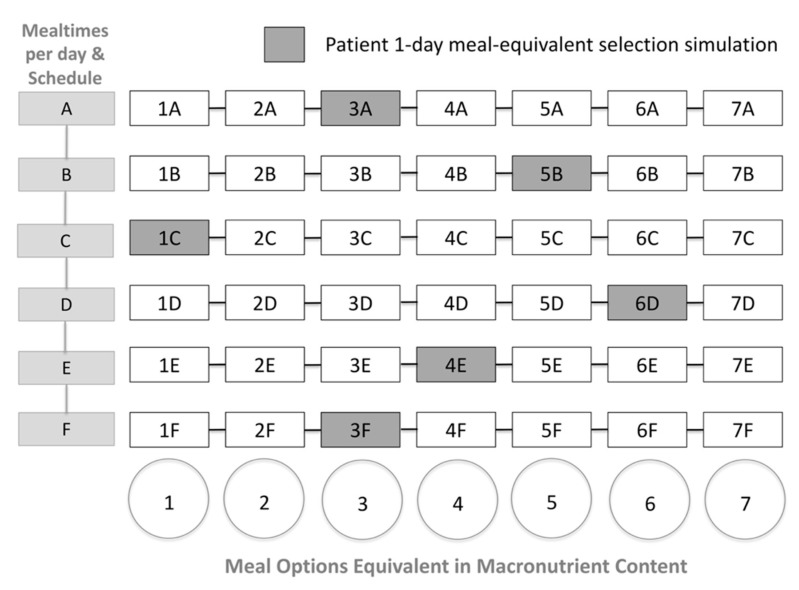
Patient 1-day meal-equivalent selection simulation exemplified in an individualized macronutrient meal-equivalent menu model.

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
