# Peer review of "Dynamic Macronutrient Meal-Equivalent Menu Method: Towards Individual Nutrition Intervention Programs"

_mps, 2019, doi:10.3390/mps2030078_

Round 1
Reviewer 1 Report
This is a well-written paper that provides good justification for personalized nutrition and for MEM, while also acknowledging the state of acceptance of personalized nutrition by professional associations.
See line 135 for a reference problem.
One of the most significant challenges for RDNs who are diet counselors is the client’s account of his/her environmentalist determinants (and a typical diet). Can you comment more on the accuracy and usefulness of the individual and environmental determinants fo the purposes of developing the MEM?
See line 201 for a grammar issue
See lines 208-209 for a grammar issue
See line 236-237 for a grammar issue
The MEM has 6 meal times... can you comment more on why that number was chosen? Meals vs. snacks?
Author Response
This is a well-written paper that provides good justification for personalized nutrition and for MEM, while also acknowledging the state of acceptance of personalized nutrition by professional associations.
See line 135 for a reference problem.
Reference corrected
One of the most significant challenges for RDNs who are diet counselors is the client’s account of his/her environmentalist determinants (and a typical diet). Can you comment more on the accuracy and usefulness of the individual and environmental determinants fo the purposes of developing the MEM?
We acknowledge this observation and include it in the conclusion of the paper.
See line 201 for a grammar issue
Grammar issue corrected: “Calculated SFS/d should be within the acceptable variation described above and…”
See lines 208-209 for a grammar issue
Grammar issue corrected: “… in accordance to the patient’s regular intake”
See line 236-237 for a grammar issue
Grammar issue corrected: “…where each column contains the different meal options, and the rows the mealtimes including the planned schedule, providing…”
The MEM has 6 meal times... can you comment more on why that number was chosen? Meals vs. snacks?
The intention of the MEM is for the RDN to establish the number and schedule of mealtimes in accordance with the patient’s characteristics and in agreement with him/her. In the manuscript this is improved in lines 214-216. Also, we modified Figure 2 to clarify this point.
Reviewer 2 Report
The manuscript needs revisions. The authors need to address the following comments.
Abstract
Clarify statement – What are the authors referring with “±”? Are these per kilogram or pound of body weight “protein ±1g/d, fat ±1g/d, carbohydrate ±2g/d and energy ±15kcal/d.”?
Introduction
Line 33 -Write down abbreviations – UN, WHO, FAO.
Lines 50-51 -This statement, “This emerging field could enable the adjustment of FBDGs for specific population needs based on their molecular profiles [6]” is not supported by reference 6. Authors need to clarify - what they are referring with “molecular profiles”?
Authors need to correct all references typos throughout the manuscript (below are some examples).
Line 89 - Correct typo “[511]”.
Line 135 - Correct typo “Error! Reference source not found.).”
Line 135 - Correct typo “[812]”, “[1216]”.
Line 149 - Correct typo “[1718]”.
Line 153 - Correct typo “[1216].
Line 155 - Correct typo [171920]”.
Line 169 - Author need to clarify why “Protein is considered the most variable macronutrient in the diet [26]”?
Line 175 - Correct typo “[2224]”.
Discussion
The discussion lack of substance, the authors need to provide sufficient information and in-depth discussion of the findings of the study with other studies available. The authors require a more critical and constructive interpretation of the literature presented in the manuscript.
Authors need to discuss - What is unique about this study compared to the literature? What gap is it filling?
It is unclear the benefits/practical implementation of the Dynamic Macronutrient Meal-Equivalent Menu Method. How many visits with the RD are needed by the patients/individuals to understand this method? What is the nutritional education and/or knowledge required by the patient/ individual to understand this method?
Lines 260-261 - It is unclear how “the MEM offers a practical approach towards individualized nutrition intervention programs.” Are there any prior studies conducted evaluating the Dynamic Macronutrient Meal-Equivalent Menu Method?
Based on the review of the literature presented, it is unclear how the MEM can improve the patients’ adherence, nutritional status, and health?
Author Response
Abstract
Clarify statement – What are the authors referring with “±”? Are these per kilogram or pound of body weight “protein ±1g/d, fat ±1g/d, carbohydrate ±2g/d and energy ±15kcal/d.”?
The symbol ± is indicative of the acceptable variations for energy and macronutrients between the theoretical calculated intake and the final result in the menu, to guarantee that the calculated standard food servings included in the nutrition plan meet the theoretical dietary intake estimates.
Introduction
Line 33 -Write down abbreviations – UN, WHO, FAO.
We used instead the complete institution names: “the United Nations Food and Agriculture Organization and World Health Organization…”
Lines 50-51 -This statement, “This emerging field could enable the adjustment of FBDGs for specific population needs based on their molecular profiles [6]” is not supported by reference 6. Authors need to clarify - what they are referring with “molecular profiles”?
Reference 6 modified to 7 and manual correction instead of using Word cross reference
Authors need to correct all references typos throughout the manuscript (below are some examples).
Line 89 - Correct typo “[511]”.
Manual correction instead of using Word cross reference [5,11]
Line 135 - Correct typo “Error! Reference source not found.).”
Eliminated from the text
Line 135 - Correct typo “[812]”, “[1216]”.
Manual correction instead of using Word cross reference [8,12] and [12,16]
Line 149 - Correct typo “[1718]”.
Manual correction instead of using Word cross reference [17-18]
Line 153 - Correct typo “[1216].
Manual correction instead of using Word cross reference [12,16]
Line 155 - Correct typo [171920]”.
Manual correction instead of using Word cross reference [17,19-20]
Line 169 - Author need to clarify why “Protein is considered the most variable macronutrient in the diet [26]”?
We modified this statement in lines 177-186 as follows: “Protein requirements depend on the individual’s health status and physical activity level, hence the wide range of protein intakes (0.75-2g/kg/BW/d). Using grams per kilogram of body weight per day (g/kgBW/d) to estimate protein intake will maintain nitrogen balance and may prevent skeletal muscle catabolism [26]. Thus we strongly recommend that protein content should be calculated prior to carbohydrates or fats, and based on g/kgBW/d instead of energy-macronutrient percentages [22,26].”
Line 175 - Correct typo “[2224]”.
Manual correction instead of using Word cross reference [22-24]
Discussion
The discussion lack of substance, the authors need to provide sufficient information and in-depth discussion of the findings of the study with other studies available. The authors require a more critical and constructive interpretation of the literature presented in the manuscript. Authors need to discuss - What is unique about this study compared to the literature? What gap is it filling? It is unclear the benefits/practical implementation of the Dynamic Macronutrient Meal-Equivalent Menu Method. How many visits with the RD are needed by the patients/individuals to understand this method? What is the nutritional education and/or knowledge required by the patient/ individual to understand this method?Lines 260-261 - It is unclear how “the MEM offers a practical approach towards individualized nutrition intervention programs.” Are there any prior studies conducted evaluating the Dynamic Macronutrient Meal-Equivalent Menu Method? Based on the review of the literature presented, it is unclear how the MEM can improve the patients’ adherence, nutritional status, and health?
There are different approaches to address menu planning based on dietary recommendations. Nevertheless, once caloric intake has been defined for the patient most methods consider the macronutrient distribution by percentage not per gram nutrient, which can lead to miscalculations and therefore limit the effect of the intervention. The manuscript discusses that when a traditional approach is used for dietary plans at 3, 6, 9 and 12 months, the non-adherence rates to the prescribed diet were 39%, 45%, 51% and 74%, respectively. The main barriers being poor self-discipline (72%), low family support (11%) and diet cost (20%).
The Dynamic Macronutrient Meal-Equivalent Menu Method presents an opportunity for HCP/RDN to accurately calculate the theoretical macronutrient dietary intake, and through a series of systematic check points during the meal-planning, be able to restrict the variation in protein (±1g/d), fat (±1g/d), carbohydrate (±2g/d) and energy (±15kcal/d) through a food-based nutritional intervention. This level of precision can be critical for patients with special nutritional requirements (sports nutrition, oncology nutrition, renal patients, diabetes, hypertension, pregnancy and lactation, etc).
Addressing the reviewer’s comment, we decided to include another sentence in the discussion section:
Lines 319-321. “Standard care refers to the minimum label of care that should be received by all patients, regardless of their nutritional status, and sometimes can include a standard diet and physical activity advice”
The proposed Dynamic Macronutrient Meal-Equivalent Menu Method has been applied in private practice (weight management, nutrition for specific activities, conditions, or pathologies, etc.). Additionally, the method had particularly good results when applied in recently diagnosed breast cancer patients in terms of adherence, and changes in fat mass and maintenance of skeletal muscle (manuscript under revision).
Reviewer 3 Report
This is a well written, interesting manuscript that will contribute to the literature.
Author Response
No comments from reviewer
Reviewer 4 Report
This paper proposed the "dynamic macronutrient meal equivalent menu method" that can prepare a nutrition plan regarding personal needs and preferences.
Some comments are as below.
1. In the methodology section, there are many narratives to describe the study's protocol; however, it's not easy to understand. I suggest that you may use flow charts with some interpretations to describe more clearly, and it can also make more readable.
2. There are many confusing and wrong citations in the text. Please correct it.
Author Response
In the methodology section, there are many narratives to describe the study's protocol; however, it's not easy to understand. I suggest that you may use flow charts with some interpretations to describe more clearly, and it can also make more readable.We included a new subsection in the manuscript entitled 4.3. Summarizing the MEM steps lines 289-311 to address your comment.
There are many confusing and wrong citations in the text. Please correct it.
All reference issues were corrected in the manuscript.
Round 2
Reviewer 2 Report
The introduction and discussion will benefit from more information from previous studies.